# An information theoretic method to resolve millisecond-scale spike timing precision in a comprehensive motor program

**Joy Ortega**[1,2], **Tobias Niebur**[3,4], **Leo Wood**[2,5], **Rachel Conn**[5,6], **Simon Sponberg**[1,2,5]*

**1** School of Biological Sciences, Georgia Institute of Technology, Atlanta, Georgia, United States of America, **2** Graduate Program in Quantitative Biosciences, Georgia Institute of Technology, Atlanta, United States of America, **3** Department of Electrical and Computer Engineering, Georgia Institute of Technology, Atlanta, Georgia, United States of America, **4** Department of Biomedical Engineering, Johns Hopkins University, Baltimore, Maryland, United States of America, **5** School of Physics, Georgia Institute of Technology, Atlanta, Georgia, United States of America, **6** Neuroscience Program, Emory University, Atlanta, Georgia, United States of America

* sponberg@gatech.edu

**Data Availability Statement:** All data presented here are available at Data Dryad: https://datadryad.org/stash/dataset/doi:10.5061/dryad.r4xgxd280. Code related to project is available on GitHub: https://github.com/AgileSystemsLab/precision The

## Abstract

Sensory inputs in nervous systems are often encoded at the millisecond scale in a precise spike timing code. There is now growing evidence in behaviors ranging from slow breathing to rapid flight for the prevalence of precise timing encoding in motor systems. Despite this, we largely do not know at what scale timing matters in these circuits due to the difficulty of recording a complete set of spike-resolved motor signals and assessing spike timing precision for encoding continuous motor signals. We also do not know if the precision scale varies depending on the functional role of different motor units. We introduce a method to estimate spike timing precision in motor circuits using continuous MI estimation at increasing levels of added uniform noise. This method can assess spike timing precision at fine scales for encoding rich motor output variation. We demonstrate the advantages of this approach compared to a previously established discrete information theoretic method of assessing spike timing precision. We use this method to analyze the precision in a nearly complete, spike resolved recording of the 10 primary wing muscles control flight in an agile hawk moth, *Manduca sexta*. Tethered moths visually tracked a robotic flower producing a range of turning (yaw) torques. We know that all 10 muscles in this motor program encode the majority of information about yaw torque in spike timings, but we do not know whether individual muscles encode motor information at different levels of precision. We demonstrate that the scale of temporal precision in all motor units in this insect flight circuit is at the sub-millisecond or millisecond-scale, with variation in precision scale present between muscle types. This method can be applied broadly to estimate spike timing precision in sensory and motor circuits in both invertebrates and vertebrates.

KSG estimation code library is available on GitHub here: https://github.com/EmoryUniversityTheoreticalBiophysics/ContinuousMIEstimation/.

**Funding:** This material is based upon work supported by NSF Graduate Research Fellowship (DGE-1650044) awarded to J.P., NSF Graduate Research Fellowship (DGE-1444932) awarded to R.C., NSF Faculty Early Career Development Award (Award no. 1554790) to S.S., a Klingenstein-Simons Fellowship Award in Neuroscience to S.S. and Air Force Office of Scientific Research Award FA9550-19-1-0396 to S.S. The funders had no role in study design, data collection and analysis, decision to publish, or preparation of the manuscript.

**Competing interests:** The authors have declared that no competing interests exist.

## Author summary

Neurons use discrete spikes of activity to encode information related to sensing and movement. The precision of the timing of these spikes is crucial for encoding this information, but how can we measure this precision? In this study, we introduce a new method for assessing timing precision by estimating the mutual information between continuous data and spike times. We do this by gradually introducing noise to the data and analyzing its impact. This approach is an improvement over previous methods that required data discretization. To test the precision, we applied this method to a comprehensive motor program that involved recording almost every spike in the primary muscles controlling the wings of an agile flying insect, a hawk moth, *Manduca sexta*. Although there were slight differences in precision among the muscles, we discovered that each muscle encoded movement information at the millisecond to sub-millisecond scale. This method can be employed to evaluate timing precision in both sensory and motor circuits across a range of animals, including insects and vertebrates.

## Introduction

Neurons in both sensory and motor systems perform temporal encoding, where information on signals is carried in the precise timing of spikes, on a faster timescale than the characteristic timescale of the signals themselves [1]. Although precise spike timing encoding in sensory systems has been found in many organisms [2, 3], the role of such temporal encoding in motor systems has been underappreciated. While strong correlations between spike rate and muscle force production exist [4–6], the precise timing of spikes in motor neurons also matters due to nonlinearities in muscle force production and mechanical interactions of musculoskeletal systems within themselves and with the environment [7]. Motor circuits in cortex [8], cerebellum [9], descending interneurons [10], and the motor periphery [11–13] carry information in neural and muscle spike timings. This information can be encoded in the difference between timings in two spike trains [11], the inter-spike interval [8], or temporal patterning [12], providing motor control systems the potential for a rich, layered code with high capacity. Neurons are capable of generating spikes with incredible precision due to biophysical properties that reduce jitter [14], yet this precision may not be consistent with the actual timescale used by the neuromuscular system for encoding and controlling movement. How precisely must the nervous system specify the timing of spikes in the motor periphery to preserve meaningful information about visual stimuli or to control the activation of muscles for movement? Knowing the scale of spike timing precision would inform our understanding of the biophysical properties of muscle and the performance constraints necessary to preserve this precision in the motor periphery, especially in motor circuits actuating different functions.

However, we have few methods to assess at what timescale information about motor output is encoded in motor circuits, and how much of the potential bandwidth of temporal codes is utilized. Previous methods for estimating precision include stimulus-response variability and reconstructions [3, 15], spike time jitter analysis in experimental and computational data [16, 17], and information theoretic methods using discrete representations of motor output and/or sensory stimuli [8, 18]. Using these methods, the firing precision of single neurons in some sensory systems has been estimated to occur on the millisecond or sub-millisecond scale [3, 16, 17, 19]. Some methods used in sensory systems do not translate well to motor systems since sensory systems can be studied using repeated or controlled stimuli, as even in well-

defined tasks like target reaching, variation exists in how a motor task is realized [20, 21]. Here information theoretic methods show promise as they do not require repeated stimuli or white noise assumptions [22, 23]. In cortical motor circuits, spike timing precision has been estimated to be at the millisecond-scale in a songbird vocalization area using a discrete information theoretic method [8]. These discrete methods, however, have drawbacks such as a limited representation of motor output.

To assess spike timing precision in motor systems, we require a method that is robust to the inherent variability in motor outputs even in response to the same sensory inputs, continuously samples this rich variability, and measures how much spiking activity in motor units encodes these outputs. To do this, we must also overcome the difficulty of obtaining spike-resolved motor unit data in a comprehensive motor circuit. Most electromyography (EMG) recordings in vertebrates either sample from too many motor units to discriminate spikes (but see [24]) or only sample single motor units with spike resolution instead of the entire motor pool, and the calcium dynamics in most imaging techniques occur over greater time-scales than the width of neural spikes. Therefore, it is difficult to compare spike timing precision in different neural circuits or muscles. Methods that both capture the features necessary to assess spike timing precision and do so in a comprehensive peripheral motor circuit would further our understanding of how the brain structures information for movement.

Hawk moths provide a compelling test-bed for questions of precision due to their fast, complex, and agile motor behaviors that are enabled by a relatively small set of muscles. One hawk moth, *Manduca sexta*, uses a set of only 10 muscles as the primary actuators of their wings [13, 25–29] and each of these muscles is innervated by one or very few motor neurons, so that each muscle is often considered a single effective motor unit [30, 31]. Therefore, we can record these 10 muscles simultaneously to obtain a nearly complete, or comprehensive, spike-resolved motor program, enabling investigation of precision over a nearly complete circuit that encodes and controls flight [13]. We know that spike timing encodes information about motor output in all muscles of this comprehensive, spike-resolved motor program [13], but do not know the scale of precision utilized for encoding and whether this scale of precision differs in different muscle types. The hawk moth motor circuit, due to its few muscles and spike resolution, enables us not only to estimate timing precision, but discover whether this precision changes depending on the specific function of each muscle. Being able to estimate precision could point to whether the nervous system encodes movement on a consistent or changing precision scale for different types of muscles.

We develop a method to assess precision down to the sub-millisecond scale in this comprehensive, spike-resolved motor program with a continuous representation of motor output in a yaw turning behavior. Our method utilizes continuous Kraskov $k$-nearest neighbors mutual information (MI) estimation [12, 13, 32, 33] with additions of uniform noise to corrupt spike timing over progressively larger time windows. We compare this method to a previously established method of estimating spike timing precision using the NSB discrete entropy estimator [8, 34, 35]. Given prior assessments of precision in motor systems and the hawk moth motor program in particular, we predict millisecond-scale precision. We test two alternative hypotheses for the precision scale across the motor program—muscles all have consistent precision scale, due to encoding a consistent amount of mutual information between spiking and the motor output [13], or they vary in precision scale. In the case where precision scale is observed to vary in different muscle types, we predict that spike timing precision may be higher in muscles with fewer spikes per wingstroke, such as the flight power muscles that have a large capacity for mechanical power modulation with small changes in phase [11].

## Materials and methods

### Comprehensive, spike-resolved motor program data set

The previously published data set used in this analysis recorded the comprehensive, spike-resolved motor program of the hawk moth, *Manduca sexta* (n = 7), and its motor output in a tethered flight preparation [13, 36] (Fig 1). Briefly, EMG signals from the 10 primary muscles actuating each moth's wings were recorded with spike-level resolution using implanted silver wire electrodes (Fig 1A and 1C). The moths were tethered using cyanoacrylate glue to a 3D-printed ABS plastic rod attached to a custom six-axis force/torque (F/T) transducer (ATI Nano17Ti, FT20157; calibrated ranges: $F_x$, $F_y$ = ±1.00 N; $F_z$ = ±1.80 N; $\tau_x$, $\tau_y$, $\tau_z$ = ±6250 mN-mm). After tethering, the moths were given thirty minutes to recover from the surgery and adapt to dark conditions, since these are crepuscular moths that typically fly at dusk and dawn. The moths were then presented with a 3D-printed plastic flower oscillating horizontally in a 1 Hz sinusoidal trajectory; these flowers have been used to elicit flight maneuvers previously and sample a wide diversity of turns (Fig 1B) [37]. The EMG recordings were sampled at 10 kHz, amplified using a 16 channel amplifier (AM Systems Inc., Model 3500), and acquired using a data acquisition board (National Instruments USB-6529 DAQ) and custom MATLAB software. The same model of DAQ board was used to acquire the strain gauge voltages from the F/T transducer used to calculate the forces and torques, also sampling at 10000 Hz.

The data set reports spike counts and spike timings in segmented wing strokes as representations of the spiking activity along with the scores of the first 2 principal components (PCs) of the within-wing stroke yaw torque ($\tau_z$) produced in wing strokes in each individual moth (Fig 1D and 1E). The first 2 PCs captured most of the variation in the original, fully dimensional representation of the within-wing stroke yaw torque; reducing dimensionality of the motor output representation makes information theoretic methods more tenable [13]. The wing strokes were segmented using a previously described method [38]. The force in the z-axis ($F_z$) was filtered with an 8-th order Butterworth bandpass filter between 5 and 35 Hz, capturing the wing beat frequency of the moth. A Hilbert transform was used to identify a common instantaneous phase, approximately at the peak downward $F_z$ in each wing stroke to serve as the zero time point, t = 0. The timing of spikes in each muscle within each wing stroke were aligned to these zero time points, such that the spiking activity were represented as continuous times in ms within each wing stroke as a matrix $S$ (Fig 1D and S1 Fig). The within-wing stroke yaw torque was also segmented into wing strokes using the identified zero time points, and a principal components analysis of the continuous yaw torque signal from t = 0 to the maximum length $L$ in samples of the shortest wing stroke of each individual moth was conducted (Fig 1E). The scores—the projections of each wing stroke onto the first two principal components—were used as a motor output representation $M$ of the within-wing stroke yaw torque, $\tau_z$. For full details on the wing stroke segmentation and PCA of the yaw torque, see the original paper [13].

### Discrete method: MI estimation using NSB

We first attempted to estimate spike timing precision using a previously described NSB entropy estimation method [8, 34, 35, 39]. As with other discrete estimators of information theoretic quantities [40], spiking activity was used to create probability distributions, $S_d$, with discrete states of spike "words" by binning the spikes in each wing stroke using different numbers of bins, $b_s$ (Fig 2A). The number of spikes that occur in each bin sets the value of that bin, and each unique sequence of bin values across all wing strokes is a spike "word", $w_i$. The prevalence of each spike word provides a discrete probability distribution, $S_d = P(w)$, that describes the spiking activity in each individual muscle. The spike timing precision, $r_d$, of each

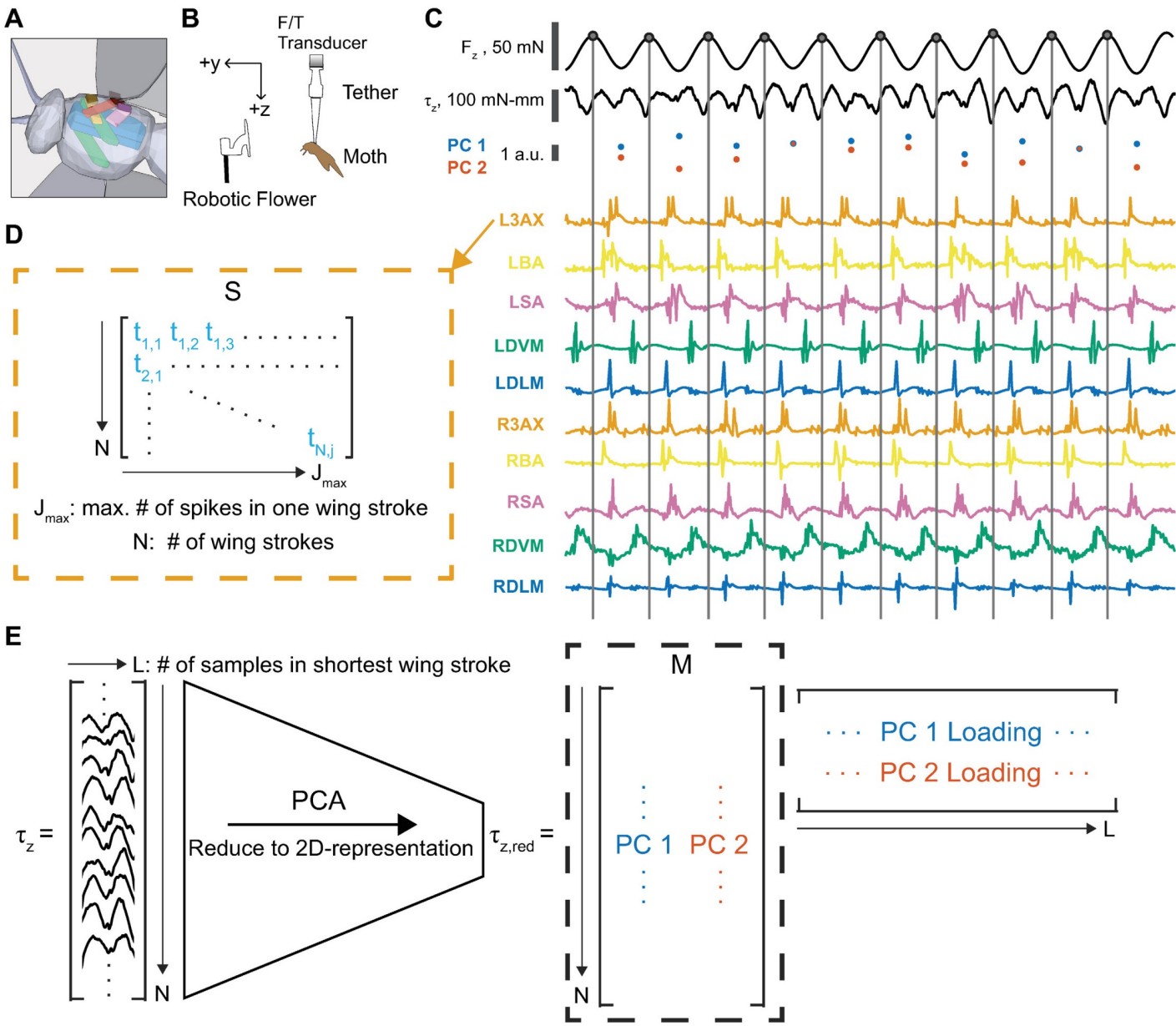

**Fig 1. Representing the spiking activity and motor output of a moth's comprehensive, spike-resolved motor program. (A)** A 3D schematic showing approximate positions and attachments of the 5 muscle types recorded in the motor program: the dorsolongitudinal muscle (DLM, blue), the dorsoventral muscle (DVM, green), the third axillary muscle (3AX, orange), the basalar muscle (BA, yellow), and the subalar muscle (SA, pink). **(B)** Tethered preparation used to record EMG signals from the 10 muscles in the motor program and the forces and torques produced as the moth responded to a flower stimulus. **(C)** Example data from 0.5 seconds of tethered flight in a moth. The bandpass-filtered $F_z$ signal is used to segment wing strokes, with t = 0 of every wing stroke corresponding to the peak downward force in $F_z$ (gray circle and line). Spike timings in each muscle and the yaw torque $\tau_z$ are aligned to t = 0 within each wing stroke. **(D)** The spiking activity $S$ for each muscle and each moth can be represented as a $N \times J_{max}$ matrix where $N$ is the number of wing strokes sampled in that moth and $J_{max}$ is the maximum spike count observed in that muscle. Where there are fewer than $J_{max}$ spikes in a wing stroke, entries with no spikes are represented as *NaN*. **(E)** The yaw torque $\tau_z$ after wing stroke segmentation can be represented as a $N \times L$ matrix where $N$ is the number of wing strokes sampled in that moth (range in our data: N = 999–2954) and $L$ is the length in samples of the shortest wing stroke in the data set, meaning that some wing strokes are shortened to length $L$ (range in our data: L = 534–668). The matrix $\tau_z$ can be reduced to a $N \times 2$ matrix using principal components analysis (PCA). The motor output $M$ is represented as the projection of each wing stroke onto the first two principal component (PC) loadings that explain the variance of the fully dimensional $\tau_z$. Data shown and panels A-B taken and adapted from Putney et al. 2019 [13, 36].

**Fig 2. Discrete Method: MI Estimation using NSB.** (**A**) Each spike train within a wing stroke is represented by the spike times represented to the sampling rate (blue). The spiking activity representation, $S$, is transformed into discrete probability distributions, $S_d$, constructed using a certain number of bins, $b$, whose entropy can be estimated using the NSB method. Each spike train in a wing stroke is binned into spike words, $w_i$, with $b$ bins. The probability distribution of occurrences of each spike word, $w_i$, is our representation of the spiking activity as a discrete probability distribution. (**B**) The motor output representation, $M$, is transformed into a discrete probability distribution, $M_d$. The scores of the first two principal components can be divided into $b_m$ partitions of equal size. In the example of this figure, $b_m = 2$ partitions produce $m_{i=1-4}$ motor output states categorized by low and high groups separated by the median score for each component. Visualization of the four motor output states, $m_i$, as a scatter plot of the scores of the first two principal components, with each unique color specifying a motor output state. (**C**) The NSB entropy estimator is used to estimate the entropies of the spiking activity $H(S_d)$ and the conditional entropies $H(S_d|M_d)$ at different numbers of bins $b$ to determine the mutual information, MI, between the spiking activity $S_d$ and the motor output $M_d$.

representation changes with the number of bins $b_s$,

$$r_d = \frac{t_{max} - t_{min}}{b}, \tag{1}$$

where $t_{max}$ is the highest spike time recorded in that muscle and $t_{min}$ is the lowest. This is the maximum range within a wing stroke where spikes occur. When this range is split into $b_s$ bins the size of each bin is $r_d$ in milliseconds.

To create a discrete probability distribution for the motor output, each column of the motor output, $M$, was partitioned into $b_m$ groups of equal size (the same number of wing strokes in each group). The case of $b_m = 2$ groups is depicted in Fig 2B; two groups for each PC score leads to a total of four discrete motor output states. For the case of $b_m = 2$, then, the motor output representation $M_d$ is the discrete probability distribution $P(m_i)$ where $m_i$ are each of the four motor output states ($i = 1$-4). In a previous implementation of this method to songbird motor cortex data, the motor output was divided into only 2 potential states using the method above [8]. Here, we use 9 total motor output states ($b_m = 3$), defining high, medium, and low values on the first two principal component axes that capture variation in the full motor output waveforms. Using 9 total states allows us to draw closer comparison with the continuous method described below, while still balancing the evident trade-off between number of motor output states and bias at high numbers of bins (S4 Fig).

Using the two discrete probability distributions of $S_d = P(w)$ and $M_d = P(m)$, we estimated the mutual information ($I_d$) between the spike "words" (represented by $S_d$) and the motor output states (represented by $M_d$) for each muscle in each moth (Fig 2C):

$$I_d = I(S_d, M_d) = H(S_d) - H(S_d | M_d), \tag{2}$$

where $I_d$ is the discrete estimate of the MI, $H(S_d)$ is the entropy of the spike word probability distribution, and $H(S_d | M_d)$ is the conditional entropy of the spike words given the state of the motor output. This conditional entropy is the sum of the entropy of the spiking activity for wing strokes in each motor state, $m_i$, weighted by the probability of that motor state, $p(m_i)$:

$$H(S_d | M_d) = \sum_{i=1}^{b_m^2} p(m_i) H(S_d | M_d = m_i), \tag{3}$$

Since all entropy estimates with finite data have bias, we used the NSB entropy estimation for the entropy of the full spike word distribution, $H(S_d)$, and the conditional entropy, $H(S_d | M_d)$, in this equation [8, 34, 35, 39], which can estimate these entropies with less bias even when severely undersampled as opposed to the more traditional direct method, or maximum likelihood, estimation [2]. The NSB method uses Bayesian estimation of the underlying probability distributions given observed events, and then estimates the first and second posterior moments of the entropies. The reported mean and standard deviation of these entropy estimates was calculated using the first and second posterior moment of the entropy, respectively [34]. Estimates of $I(S_d, M_d)$ were done for spike word distributions, $S_d$, defined with increasing numbers of bins $b_s = 1 - 70$, with $b_s = 70$ being within the sub-millisecond range in all muscles. With this method, if a plateau in MI can be identified as $b_s$ increases, a breakpoint analysis or other algorithm can be used to determine the threshold value of spike timing precision, $r_d$, as the point where MI begins to fall with decreasing numbers of bins $b_s$.

While the NSB entropy estimator typically performs much better than traditional maximum likelihood estimation at reducing systematic bias in undersampled regimes, it is not a bias-free estimator. As the dimensionality in the motor space (number of motor output states $b_m^2$ in $M_d$) and the spike word distributions (number of bins $b_s$ in $S_d$) is increased, the

estimated MI will increase (S4 Fig). A consequence of this is that often MI estimated at higher numbers of bins $b_s$ will continually increase due to bias, never reaching a plateau and thus preventing identification of critical value for spike timing precision. To counteract this, we used a simple shuffling bias correction similar to [41, 42], where an estimate of bias, $I_{sh}$, was calculated as the mean of many repeated conditional entropies $H_{sh}(S_d|M_{d,sh})$ with motor output states randomly shuffled relative to the spike word distributions. The mean MI from many repeated shufflings $I_{sh}$ was then subtracted from the original MI estimation $I_d$, giving a shuffling bias-corrected MI

$$I_{d,sh} = I_d - I_{sh} = (H(S_d) - H(S_d|M_d)) - (H(S_d) - \frac{1}{n_{sh}}\sum_{i=1}^{n_{sh}}H_{sh}(S_d|M_{d,sh})) \qquad (4)$$

where $n_{sh}$ is the number of repeated shufflings performed (typically $n_{sh} \geq 10$) and $M_{d,sh}$ are the motor output states randomly permuted relative to associated spike words. With this shuffling bias-corrected MI, the threshold value of spike timing precision is estimated as the number of bins $b_s$ where MI $I_{d,sh}$ peaks before falling as bias begins to increase, rather than where MI plateaus. Identifying the precision of spike timing is easier as a peak-finding rather than plateau-breakpoint problem. It also allows more fine-grained discretization of motor output, which typically leads to runaway bias at high numbers of bins to not prevent estimation of spike timing precision.

## Continuous method: MI estimation using KSG and added noise

Our second method of estimating the spike timing precision used a continuous Kraskov $k$-nearest neighbors MI estimation (KSG) [32] (Fig 3). This method improves on the discrete method outlined above by using a MI estimator which operates on a continuous representation of both the spiking activity and motor output, estimating entropies from statistics of the $k$-nearest neighbor distance distributions of the data. Using a continuous estimator allows for testing of spike timing precision by addition of uniform noise, which grants finer control then when limited to specific bin sizes.

For the spike timings in each wing stroke, we generated added noise from a uniform distribution, $U$, between values of 0 and $r_c$, where $r_c$ set the width of the noise window (Fig 3A). This shifts spike timings in $S$ by $U(0, r_c)$. Our spike timing representation, then, is a matrix of corrupted spike timings $S' = S + U(0, r_c)$ of size $N \times S_{c,max}$ where $N$ is the number of wing strokes and $S_{c,max}$ is the maximum number of spikes observed in a wing stroke for that muscle in each moth (Fig 3B). Our continuous estimate of MI, $I_c$, is then estimated between these two quantities $S'$ and $M$ using the KSG estimator, which uses the Euclidean distances between each sampled wing stroke and its $k^{th}$ nearest neighbor in a space defined by the variables $S'$ and $M$ to estimate the joint entropy of the two variables $H(S', M)$ and the mutual information between the two variables $I(S', M)$. The continuous KSG estimation of the mutual information $I_c = I(S', M)$ across values of $r_c$ takes the form:

$$I_c = I(S'_c; M) + \sum_{i=1}^{S'_{c,max}}p(S'_c = i)I(S'_t; M|S'_c = i) \qquad (5)$$

where $S'_t$ is the set of spike times in $S'$, $S'_c$ is the spike count, and $S_{c,max}$ is the max count of spikes in a single wing stroke observed for each muscle. MI between spike timings with added noise $S'_t$ and motor output $M$ are calculated separately for each possible number of spikes $i = 1, \ldots, S'_{c,max}$ in a wing stroke and summed weighted by the probability of a wing stroke having that many spikes, $p(S'_c = i)$. Calculation of $I_c$ in this manner separates spike count MI $I(S'_c; M)$

## Continuous Method: MI Estimation using KSG and Added Noise

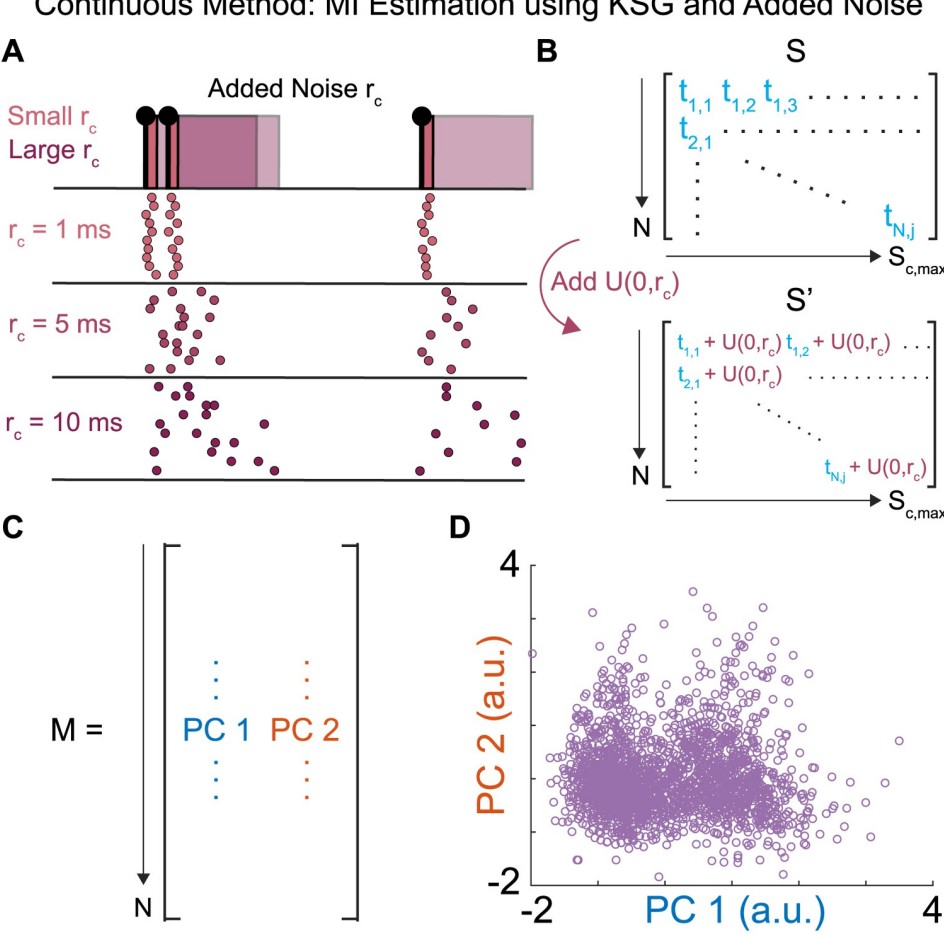

**Fig 3. Continuous Method: MI Estimation using KSG and Added Noise.** (**A**) Uniform noise, $U(0, r_c)$, is added to the precise spike timings. At small $r_c$, the noise corruption does not change the spike train representation much over 10 iterations. At large $r_c$, significant variation between the representation of the same spike train appears over 10 iterations. (**B**) The representation of the spiking activity, $S$, has a uniform window of noise defined by width, $r_c$, added to it to create representations, $S'$, at varying levels of added noise. (**C**) The motor representation for the continuous method, a $N$ x 2 matrix of the scores of the first two principal components for each wing stroke. (**D**) Visualization of $M$ as a scatter plot of the scores of the first two principal components, where each point is a wing stroke.

from spike timing MI by conditioning spike timing on spike count [12, 13]. The mutual information between the spike timings with added noise $S'$ and the scores of the first 2 PCs in each wing stroke $M$ was estimated 150 times with different random noise samples for each value of $r_c$ tested, reducing the effect any individual estimation run has on the final mutual information value. Therefore, we report the mean and standard deviation of these 150 estimates of $I_c$. MI estimates with no added noise were previously assessed to be stable on this dataset by using data fractioning and by varying values of $k$ [13]. The KSG estimator was used with $k = 4$ throughout this work, though estimated precision was found to not be sensitive to the choice of $k$ up to $k = 7$ (S7 Fig).

To determine a precision value for each individual muscle, we used the error estimate of the MI at $r_c = 0$ to set a range of expected values. We then defined the precision for a muscle as the point where the mean MI for all 150 estimates falls below the lower bound on the estimate of the MI (the mean minus the standard deviation) at $r_c = 0.0$ ms (i.e. the original data) based

on variance in data fractions [13, 33]. We determined the standard deviation of our MI estimates at $r_c = 0$ by subsampling the data sets in non-overlapping data fractions. The variance in these fractions was used to estimate the standard deviation or uncertainty of the estimate at the full data size, as previously [33]. To test for statistically significant differences between the spike timing precision $r_c$ of different muscles, we used one-way and two-way ANOVA tests, as well as the non-parametric Kruskal-Wallis tests.

We also tested two alternate methods for choosing the spike timing precision $r_c$ (S5 Fig). The first of these, a derivative method, estimated spike timing precision $r_c$ as the noise level where the 2$^{nd}$ derivative of $I_c$ with respect to the noise level $r_c$ peaks. The second method, a line fitting intersection approach, treats the $I_c$ vs. $r_c$ curve similar to a phase transition, fitting two lines to the approximately linear low-noise and high-noise regions at either end of the curve and estimating spike timing precision as the noise level where these two lines intersect. All three algorithms produced similar results on real and simulated data with precision fixed to known levels (see S5 Fig), with the standard deviation method used throughout this work chosen for its reduced systematic bias and simplicity. While the three methods did give slightly different precision estimates and displayed different bias characteristics, all three estimated precision within the range of 0.5–3 milliseconds on the hawk moth comprehensive motor program data, indicating relative robustness to the specific procedure used to choose the noise level at which MI falls.

## Results and discussion

### The continuous method gives an improved estimate of millisecond-scale timing precision

Here, we present an extension of a continuous MI estimation method [33] to obtain reliable estimates of millisecond-scale timing precision (Fig 4) across the hawk moth's comprehensive motor program. The discrete method, which has been previously used to show spike timing precision in motor systems [8, 34], is able to estimate spike timing precision and demonstrate that each muscle from each moth exhibits higher precision than a spike count code.

One of the advantages of the NSB estimator over other discrete estimators is that it can estimate probability distributions from very sparse sampling, though in our data set strong under-sampling bias at high numbers of bins necessitated shuffling bias correction to produce spike timing precision estimates (Fig 4A–4C and S4 Fig). Even with bias correction and more fine-grained motor output states, estimates are not smooth with $r_d$, with large jumps in $I_d$ between adjacent values of $r_d$ which make determining a specific peak MI more difficult. Across all muscles except the dorsolongitudinal muscles (DLMs), the discrete method found significantly lower precision than the continuous method, indicating that the continuous method is more capable of detecting associations between spike timing and motor output on smaller time-scales. Similar to comparisons between continuous and discrete estimators of transfer entropy on spike data [40], the continuous method with additive noise allowed determination of precision at a finer resolution than the discrete method, with greater accuracy and reduced bias (Fig 4A–4C).

Precision estimation with the continuous method is still limited; precision not only cannot be smaller than the minimum time representation of 0.1ms defined by the 10kHz sampling rate, but also the Kraskov estimator operates at a resolution determined by the k$^{th}$ nearest neighbor distance, bounding the resolution of entropy estimation by the size of the dataset used. With more wing strokes, the space of data would be denser and thus typically have a lower k$^{th}$ nearest neighbor distance. These limitations, however, merely indicate that the true value of precision may be lower than that identified by the continuous method. For this

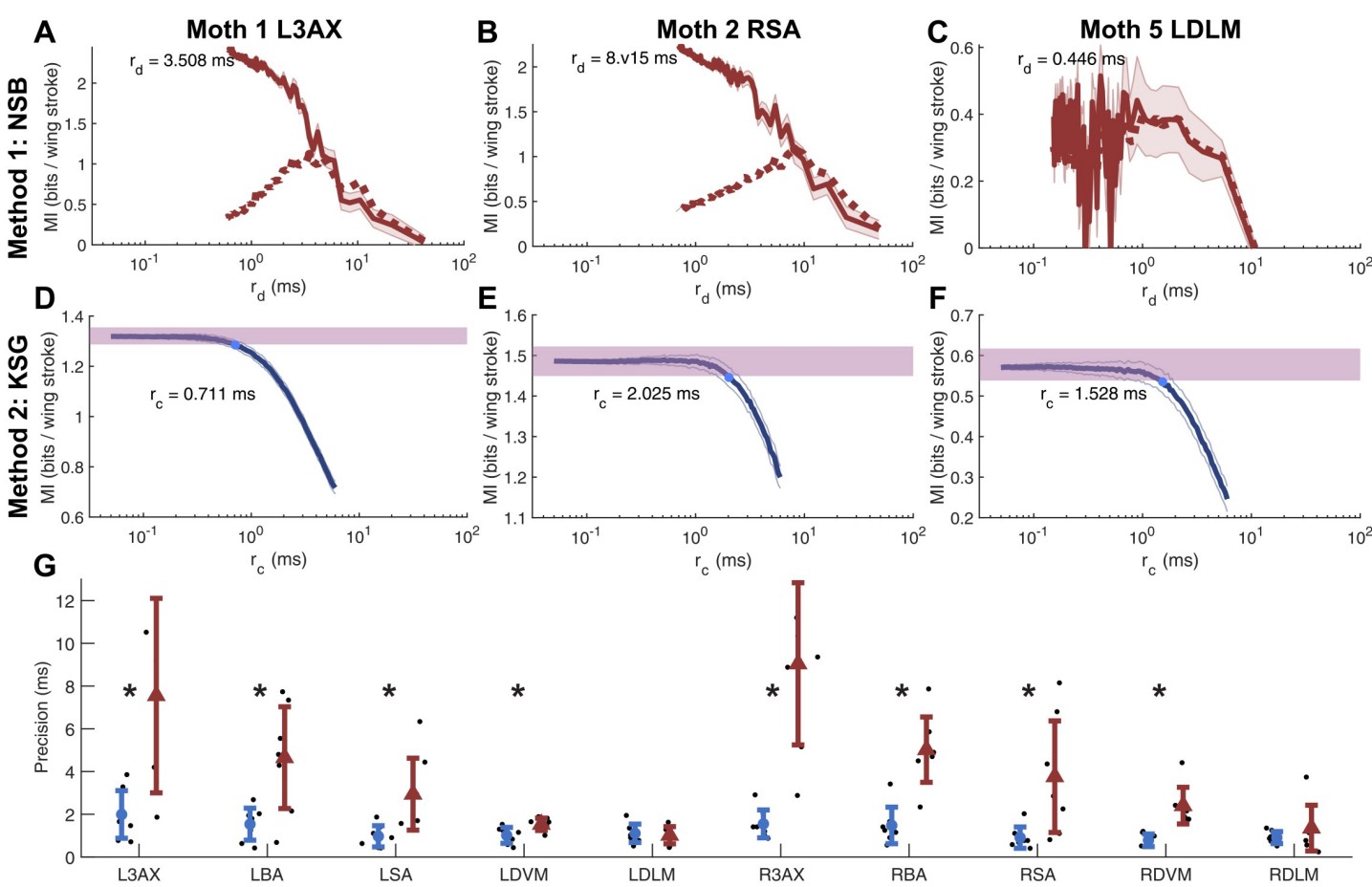

**Fig 4. Comparing the two methods used to estimate spike timing precision. (A-C)** MI $I_d$ (solid line, mean ± STD) and shuffling bias corrected MI $I_{d,sh}$ (dotted line) from the discrete NSB method for three example moth, muscle pairs as bin size $r_d$ is varied. Markers and text denote estimated spike timing precision. **(D-F)** MI, $I_c$, (mean ± STD of 150 tests of adding uniform noise) from the continuous method for the same three example moth, muscle pairs as uniform noise amplitude $r_c$ is varied. Purple area denotes ± one STD error estimate of MI at zero noise, markers and text denote estimated spike timing precision. KSG estimator run with $k = 4$ for this and all main text results. **(G)** Mean ± STD of estimated spike timing precision for continuous method (blue circles) and discrete method (red triangles) for all muscles. Black asterisks indicate when spike timing precision estimates for a muscle are significantly different between the two methods according to a two-sample t-test ($p < 0.05$).

dataset, too, the introduction of uniform noise does not dramatically affect the scale at which entropy is estimated by changing the max-norm k$^{th}$ nearest neighbor distance distributions (S2 Text, S6 Fig), nor does changing the operating scale by changing $k$ significantly alter observed precision values (S7 Fig).

## Validation on simulated data

To better understand the performance difference between the two methods and validate their ability to accurately identify precision, we created data with known precision (Fig 5).

Both estimators were repeatedly run on either the real data from the comprehensive hawk moth motor program or on a synthetic dataset with spike timing precision *a priori* fixed to known resolution. By rounding spike times to the nearest multiple of a desired resolution (Fig 5A), we observed the error and biases in each method at estimating spike timing precision.

The continuous noise addition method outperformed the discrete method at accurately and consistently estimating the precision of both datasets (Fig 5E and 5G). For all hawk moth data and synthetic data, the continuous method produces a smooth monotonic decrease in MI

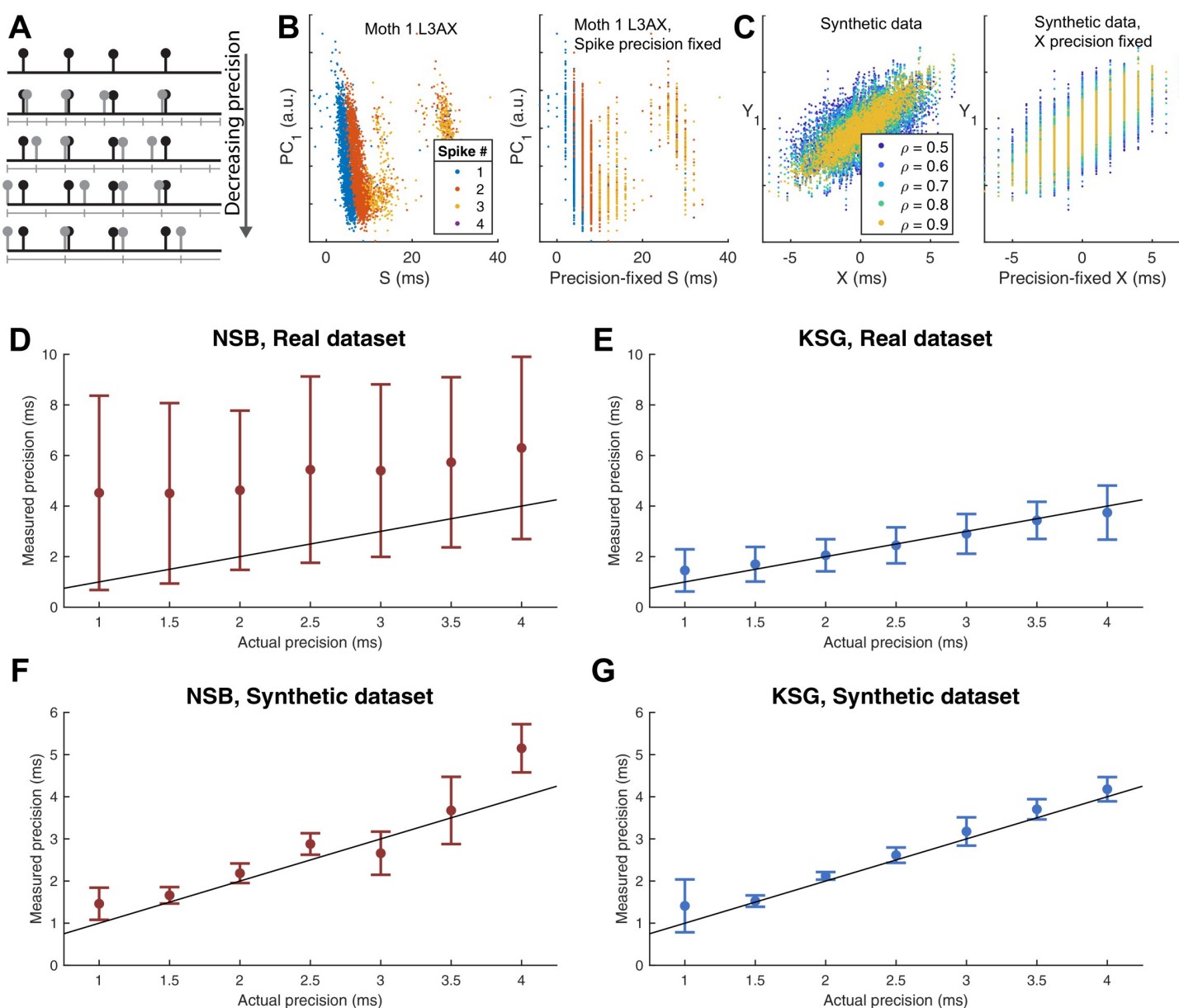

**Fig 5. Performance of continuous and discrete methods at estimating precision of different types of data fixed to known precision levels. (A)** Spike timings of any dataset can be fixed to a desired ground-truth precision by rounding to the nearest multiple of a desired resolution. **(B)** Example of fixing the precision of spike time data from left axillary data of moth 1. **(C)** Example of synthetic data being fixed to specific precision. Synthetic data is composed of a single $1 \times N$ vector of "spike times" related to a $2 \times N$ matrix of "motor PC scores" via bivariate gaussians ($\mu = 0$, $\sigma = 2$) with correlation $\rho$ varying from 0.5 to 0.9. Each synthetic trial consisted of $N = 2500$, with precision estimation at each correlation level repeated 4 times. **(D)** Mean ± STD of spike timing precision estimated by NSB discrete method, with shuffling bias correction, against actual precision for hawk moth motor program data fixed to specific precision levels. Motor outputs were discretized to $b_m = 3$ states (9 total states). Black unit line in all plots indicates perfect estimation, where estimated precision directly matches actual precision. **(E)** Mean ± STD of spike timing precision estimated by KSG continuous method against actual precision for hawk moth motor program data fixed to specific precision levels. KSG estimator was run with the same parameters as the rest of the text. **(F)** Mean ± STD of spike timing precision estimated by discrete method against prior fixed precision for synthetic data. **(G)** Mean ± STD of spike timing precision estimated by continuous method against prior fixed precision for synthetic data.

with increasing noise (Fig 4D–4F and S2 Fig), as compared to the relatively jagged and unstable estimates from even the bias-corrected NSB method (Fig 4C and S3 Fig). The continuous method had lower uncertainty than the discrete method and demonstrated little systematic tendency away from the true precision value, other than a tendency to estimate less spike

timing precision at low actual values of precision around 1 millisecond or less (Fig 5G). This indicates that at sub-millisecond scales, the continuous estimator is consistently conservative, with actual values potentially more precise than reported. The discrete method, meanwhile, has larger error and variation in precision estimates, particularly for the higher dimensional real data (Fig 5D and 5F). While the discrete method rarely estimated precision to be lower than its actual value, meaning it can potentially be used to at least bound the true precision, it had high variance in its estimates which makes it less reliable for estimating spike timing precision.

Part of what hinders the discrete method may be the required discretization of motor output states. Our implementation of the discrete method estimated mutual information for a motor output probability distribution $M_d$ that included more motor output states $m_i$ (9 total states) than its previous implementation [8]. We were only able to achieve this more fine-grained resolution in motor output by implementing shuffling bias correction, as more motor output states leads to greater undersampling bias (S4 Fig). A previous implementation of NSB was able to robustly identify spike timing precision down to 1 ms in a songbird motor cortex analog by averaging mutual information estimates across all neurons sampled [8], but was unable to assess the precision of individual neurons and if there were differences in precision scale across the population recorded. Our introduction of shuffling bias correction to NSB does allow for more fine-grained resolution in motor output by counteracting the greater undersampling bias seen with more motor output states (S4 Fig). But even with bias correction it is clear the discrete method fails to capture smaller resolution features of the data at the available sample sizes.

With the continuous method, the KSG estimator operates at a scale which varies with the sample size and distribution of the data [32], producing a much richer representation of the spike and motor output spaces without having to trade off for significant bias. Additionally, the continuous method has the advantage of being able to more finely represent different levels of spike timing precision, especially at small values of $r_c$, by relying on uniform noise rather than binning spike trains. For data with a large range of spike counts, the number of bins required by the discrete method to represent spike timing precision of 1ms or less can become too large for the method at the available sample sizes, leading to undersampling bias which makes discrimination of the actual information limit difficult. With the continuous method, we do not have the problem of balancing between more bins and greater bias, as the input spike time data is always approximately the same, only with values which are corrupted by noise. This enables us to determine a specific estimate of spike timing precision at far finer scales with the continuous method.

In both the KSG and NSB estimates presented here, we have reduced the motor output state representation from the fully dimensional torque waveforms in each wing stroke. In the motor output representation utilized in the NSB method, we have discretized to nine distinct motor output states. Even in the KSG method, we only represent the motor output as the scores on those first two principal component axes. It is possible that variation captured in other principal component axes would affect our precision values, but by increasing the dimensionality of our motor output, we would also sacrifice stability and introduce bias into our mutual information estimates for both methods. Previously, we demonstrated that the KSG MI estimates with no noise added were robust to changing the number of principal components included [13], and we used two principal components here to keep our results consistent with our previous work.

Finally, it is important to note that both methods find spike timing precision from the shape, not magnitude, of MI against bin size $r_d$ or noise amplitude $r_c$. Here, we are able to distinctly consider information that is temporally encoded, without also including information

that might be preserved in a rate (count per wing stroke) code [1]. One important consideration when using KSG MI estimation is to ensure that estimates are stable in smaller data fractions and robust to the choice of the free parameter, $k$ [33]. This also extends to the estimation of spike timing precision using KSG; while we used $k = 4$, the choice of $k$ within a reasonable range should not affect precision estimates. This data set has already been checked for stability and robustness to choice of $k$ and smaller data fractions [13], as well as the robustness of precision estimates to choice of $k$ up to $k = 7$ (S7 Fig).

Other methods besides those presented here have been used to estimate spike timing precision. For example, many experimental studies in sensory systems have investigated the statistics of spike timing jitter as a proxy for how precise the neural system must be to encode information about a repeated stimulus [15, 18, 43–45]. Computational studies have also used spike timing jitter to measure the reliability of neural coding under repeated presentations of the same sensory stimulus [16, 46]. In motor systems, however, we cannot obtain many realizations of the same motor output to investigate jitter in spike timings because it is difficult to constrain motor behavior. Even in well-defined tasks like target reaching, variation exists in how a motor task is realized [20, 21], so any spike timing jitter may be encoding this variation and not representative of the true precision of the spiking activity. Another method of estimating spike timing precision used added Gaussian noise windows in data from model LGN neurons to estimate mutual information between spike trains and visual stimulus movies [17]. Here, instead of Gaussian noise, our method uses uniform noise which corrupts information evenly across the window, $r_c$, which we use to define the spike timing precision. Our method of noise addition preserves less information about when the spike occurred, whereas adding a Gaussian window of noise will still preserve that information because the distribution of jittered spike times will peak at the original spike timing. The continuous method developed here can be used for both sensory and motor systems to determine the scale of spike timing precision at fine resolution in data sets with rich variation either in sensory inputs or motor outputs.

While we have known that computational models of noisy neurons in Kilinc et al. can demonstrate precisely timed spikes [46] and neural networks can learn precisely timed sequences of spikes [47], we can now assess the degree of spike precision across the entire comprehensive motor program in individual motor units. Methods like spike distance metrics have been used to assess spike timing precision in populations of neurons and neurons [48, 49], but with our method we can compare the scale of spike timing precision in individual units in a population. We can now construct a picture of how individual neurons in a comprehensive motor circuit use precise spike timing to control movement.

## Every muscle in the hawk moth flight motor system uses millisecond-scale spike timing precision

Using the continuous method, we can reasonably assess the spike timing precision across a comprehensive motor program and probe whether motor units filling different functional roles encode information on the same or different precision scales. We found spike timing precision to the millisecond or sub-millisecond scale across all ten muscles in the motor system as well as statistically significant differences in the spike timing precision of functionally distinct muscle types (Fig 6A). When left and right muscles are considered separately, there are marginally statistically significant differences across the muscle groups, likely due to asymmetries in tethering. However, all mean values across individuals and muscles range from 0.8 to 2.0 ms. We combine left and right muscles of each type to determine whether different muscles have statistically different spike timing precision values (Fig 6B).

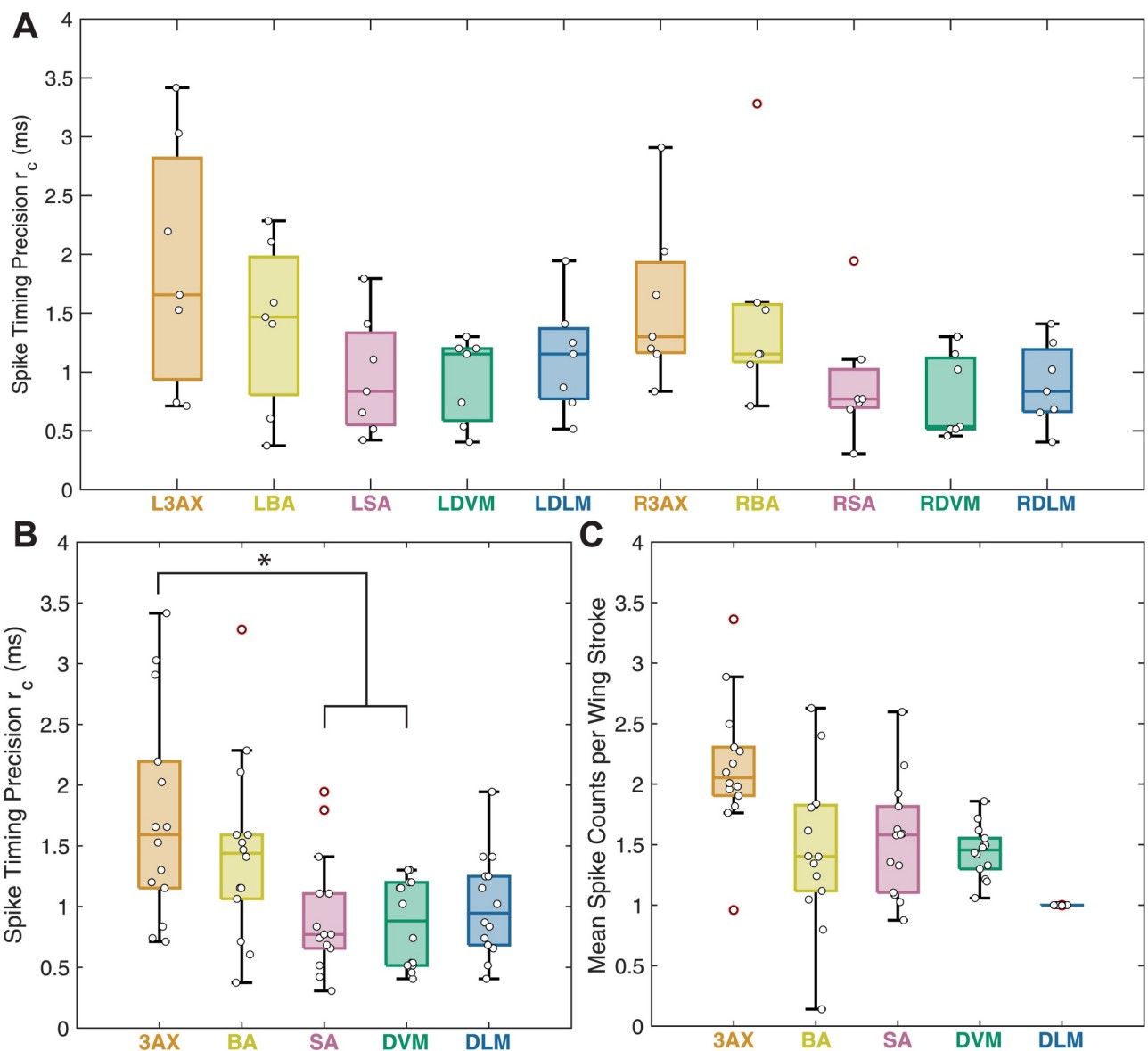

**Fig 6. Spike timing precision estimates from the Continuous Method using the lower bound choice of precision value, $r_c$.** (A) Spike timing precision values ($r_c$) for all moths reported as boxplots (middle black line: median; boxes: 25th and 75th percentiles; whiskers: all data except outliers; white circles: $r_c$ values for individual moths; red circles: $r_c$ values considered statistical outliers). Marginally statistically significant differences were found across muscle types (one-way ANOVA, $p = 0.0115$; Kruskal-Wallis, $p = 0.0481$). (B) Spike timing precision values ($r_c$) for all moths with left and right side muscles combined (same data as A). The asterisk denotes groups with statistically different means in a Kruskal-Wallis non-parametric test. The precision values of the 3AX muscle were significantly different from the SA and DVM in a multi-comparison Kruskal-Wallis test at significance level $p < 0.05$. (C) Mean spike counts for all moths with left and right muscles combined as in B. The asterisk denotes groups with statistically different means in a Kruskal-Wallis non-parametric test. The spike count means of the DLM was significantly different from all other muscles in a multi-comparison Kruskal-Wallis test at the same significance criteria as in B ($p < 10^{-6}$). The 3AX muscle is significantly different from all other muscles and the DLM is significantly different from the SA muscle in a multi-comparison two-way ANOVA with the same significance criterion (two-way ANOVA for muscle type, $p < 10^{-6}$; and left/right sides of the moth, $p = 0.62$; with a test for interaction, $p = 0.93$).

Each of the five left-right muscle pairs in the hawk moth motor program—the dorsolongitudinal (DLM), dorsoventral (DVM), third axillary (3AX), basalar (BA), and subalar (SA) muscles (Fig 1A and 1C)—have different sizes, different attachment points, different activation patterns, and different biomechanical roles. The DLM and DVM indirectly actuate the

downstroke and upstroke of the wing, respectively, by deforming the thorax. The 3AX, BA, and SA muscles directly attach to the base of the wing to fine-tune the motion of the wing during flight. The DLM and DVM have been traditionally thought of as flight power muscles, while the 3AX, BA, and SA have been called flight steering muscles [26]. These two functional roles could require different spike timing precision scales. Most of these muscles spike several times per wing stroke, but the DLMs only spike once, meaning that any encoded information must be present in spike timings [11], perhaps indicating that they require higher precision.

However, in all muscles, not just the DLMs, spike timing is important for motor output [13]. Sub-millisecond scale timing in the DLM is linked to power output during flight [11, 50]. Additionally, changes in the spike timing of the DVM and 3AX in a sister species of hawk moth to the one we investigate here have been shown to correlate with wing kinematic changes at wing stroke reversal [51]. In *Manduca sexta*, changes in timing of the BA is correlated with turning behavior and changes in the timing of the SA is correlated with wing depression, promotion, and remotion [26, 27]. Because each of these muscles is functionally distinct, we investigated whether the scale of spike timing precision differed by muscle type.

We estimated the highest spike timing precision ($r_c$) in the subalar (SA) and dorsoventral (DVM) muscles, with mean precision values of 0.92 ms and 0.85 ms for the SA and DVMs, respectively, across all moths, making them on average sub-millisecond precise. We also found the DLM muscles to be sub-millisecond precise on average. Previously, bilateral timing differences between the left and right DLMs were shown causally to have sub-millisecond precision for producing yaw torque [11]. Even in an asynchronous flier, which flies at high wing beat frequencies where motor neurons innervating the DLMs do not fire within every wing stroke, bilateral calcium modulation in the DLMs from these motor neurons plays a role in controlling the mechanical power produced in turning flight [52]. We can now demonstrate that both sets of flight power muscles, the DLMs and DVMs, encode information on the sub-millisecond scale as individual muscles, not just in relation to each other. The ability to fine tune power production during agile flight is driven by sub-millisecond changes in these power muscles, and sub-millisecond differences in their timings relative to each other. Evidence of sub-millisecond scale encoding in the SA muscles points to a precise control role for these muscles, perhaps in controlling wing depression and remotion at specific wing stroke phases [26].

The 3rd axillary (3AX) muscles have the lowest estimated spike timing precision at 1.78 ms (mean across all moths) and are statistically different from the SA and DVM muscles. Since these muscles have previously been shown to encode about the same amount of spike timing information [13], this suggest that there is no relationship between the precision of spike timing and the amount of encoded information. While all these muscles encode approximately the same amount of spike timing information, they have different overall amounts of variation in spike timing during a wing stroke (S1 Fig). The 3AX muscle is able to encode the same amount of information with less spike timing precision than other muscles. While not statistically significant in all statistical tests, the mean spike count of the 3AX muscle in each wing stroke is higher than all other muscle types in the motor program (Fig 6C). It could be that having multiple spikes to carry motor information decreases precision requirements for each spike. The DLM, for instance, is one of the most precise muscles in the motor program, which may be driven by the constraint of having only one spike with which to encode information. One confound to this observation is that muscles with more spikes per wingstroke provide higher dimensionality to the MI estimator, which leads to greater max-norm nearest neighbor distances (and thus potentially lower precision) for the same sample size. However, for the sample sizes and spike counts explored in this paper, we found that this is not a major confound to the estimated precision levels (S2 Text, S6 Fig). In the simulated data, precision is estimated within ±0.5 ms of ground-truth precision level even for muscles of high dimensionality

such as the 3AX and BA (Fig 5). Similarly, for the same data, intentionally increasing nearest neighbor distances by choosing $k > 4$ does not alter precision estimates (S2 Text, S7 Fig).

We also used two alternate methods to estimate spike timing precision for each muscle, which involved either peak derivatives or the intersection of two lines fit to approximately linear regions of the $I_c$ vs $r_c$ curve. Our findings about the precision of these muscles is robust to changing the method of choosing a precision value (S1 Text, S5 Fig). With all methods, all muscles are found to be precise on the same <2 millisecond time scale, and the 3AX and SA are still found to be the least and most precise muscles, respectively. The derivative method tended to estimate more muscles as sub-millisecond precise, but simulated data indicated a tendency to estimate lower than actual precision and it only significantly differed from the main method of this paper on the 3AX and BA muscles. The line intersection method gave significantly different estimates for the SA, DVM, and DLM as precise to a 1–1.5 millisecond time scale, but even this only differed from the other methods by on average 0.5 ms. All methods of choosing the spike timing precision are viable, but we report more substantively on the method that explicitly incorporates the lower bound of our MI estimate at $r_c = 0$, since it accounts for uncertainty in our information estimation, has favorable accuracy, and is the simplest to implement.

The spike timing precision of individual muscles may also enable coordination between muscles. The studies cited above that demonstrate the importance of timing throughout the hawk moth motor program all reported spike timings of muscles relative to the DLM [26, 27, 50, 51], and the study on the DLM timing investigated the relative timing between the left and right DLMs [11]. Here, we demonstrate that all individual muscles have precision on the sub-millisecond to millisecond-scale, which could provide the precision necessary for this type of coordination across muscles.

## Spike timing must be essential for sensorimotor integration

Because millisecond-level spike timing precision is present across the entire motor program, not just in specialized muscles, encoding of this precision must either 1) be preserved throughout the nervous system from sensory inputs to interneurons then out to the periphery, or 2) be accomplished in the periphery through local sensory inputs to motor neurons and transformation of descending commands to a more precise timescale. Many sensory systems are millisecond-scale precise [3], much like the flight motor program investigated here, so it is possible that millisecond precise spike timing could be preserved in transformations of activity throughout the central nervous system to muscles. In the hawk moth investigated here, *M. sexta*, it is known that visual-responsive dopaminergic interneurons heavily innervate the pterothoracic ganglion with axon branches where the motor neurons controlling the five muscles studied here originate [53, 54]. These interneurons could preserve temporally precise information about the visual scene that is passed through synapses to produce precise spiking. Dopaminergic interneurons that synapse directly with the motor neuron for the basalar muscle (b1) in flies enable wing coordination during onset and termination of flight by integrating bilateral sensory inputs [55]. Additionally, precise timing in a GF interneuron helps determine action selection in a *Drosophila* escape response [10]. This demonstrates the importance of precise timing in the central nervous system for action selection and may point to the preservation of precision throughout a sensorimotor circuit since these GF interneurons receive input from the optic lobes.

An alternative possibility is that precision arises not from precise visual encoding preserved through descending interneurons, but from sensorimotor connections in the periphery from mechanosensors. In flies, activation of haltere steering motor neurons is

known to modulate the spike timing of a basalar muscle on the millisecond-scale [56], facilitated by a direct, electrotonic connection between haltere afferents and the motor neuron of that muscle [57]. Haltere mechanosensors are typically campaniform sensilla, which have been shown to encode lateral displacements using precise spike timing [58]. While moths do not have halteres, they do have campaniform sensilla on their wings that respond within milliseconds to mechanical stimulation [59]. These temporally precise sensors could dictate temporally precise responses in muscles [60, 61]. Flight posture reflexes have been activated by stimulating these wing mechanosensors [62], which may indicate the presence of analogous direct monosynaptic connections between wing mechanosensors and motor neurons in the hawk moth. It is also possible that both these peripheral sensorimotor connections as well as descending inputs carrying visual information play a role in preserving spike timing precision in the motor periphery that is necessary for the robust, agile execution of flight.

The presence of highly temporally precise encoding is not a special case reserved for certain functionalities, but rather a pervasive encoding strategy used by the nervous system. While previously sub-millisecond precision had only been demonstrated in a bilateral pair of muscles in the hawk moth flight motor program, we now have shown at least millisecond-level encoding precision in each individual muscle that the moth uses to control its wings during flight. Precision is likely necessary due to large changes in power output driven by millisecond-scale changes, which belies the assumption that muscle can be treated merely as a low pass filter on motor neuron activity, where spike rate is proportional to force produced. Millisecond changes in spike triplets in motor units of song bird breathing muscles causally change pressure production in their lungs [12]. The biomechanical and molecular properties of muscle change throughout their cycle of activation, and the millisecond scale timing of activation during these state changes could cause muscle to produce different forces [7]. For example, identical spike triplets in vertebrate muscle can produce different force outputs depending on whether they occur at the onset of or during tetanus [63].

Therefore, spike timing precision is likely not a special case for invertebrates or for fast behaviors like insect flight, and should be investigated in vertebrates and other motor behaviors. Hawk moth flight is only intermediately fast, with wing strokes elapsing 40 to 50 milliseconds, temporal encoding should not be assumed to only occur in insects or other animals with fast frequency behaviors. Walking or running can be a much lower frequency behavior, but bipedal foot strikes—which occurs on a similar timescale to a moth wingstroke—may use precisely timed motor signals [64]. Slower time scale behaviors, like breathing in songbirds, have also been shown to encode on the millisecond scale [12]. Other motor behaviors in vertebrates also can occur on fast timescales, like eye movements, finger snapping, and typing. Additionally, neural mechanisms in vertebrates could be used to encode and preserve spike timing precision across synapses. As an example, stretch reflexes use direct sensory-to-motor connections that are analogous to monosynaptic stretch reflexes in invertebrates mediated by chordotonal organs [65], and may serve as sources of precision in vertebrate motor units. The spike timing precision of motor neuron activity in a variety of behaviors and species can be assessed using the method presented here, which provides a specific estimate of the scale of precision for encoding a motor output.

## Supporting information

**S1 Fig. Raster plots from five muscles along with corresponding PC scores.** Timing of the L3AX, LBA, LSA, LDVM, and LDLM muscle spikes for 200 example wing strokes in an individual moth (scale bar = 20.0 ms). Spikes are color-coded by their order in each wing stroke.

The corresponding values in arbitrary units of the PC scores in each wing stroke are displayed in the far right panel.
(EPS)

**S2 Fig. Estimates of $I_c$ from the continuous KSG method for all moth and muscle pairs.** Log plots of $I_c$ at different values of $r_c$ for all moths and muscle types. Markers indicate precision values for each $I_c$ curve found using the zero-noise standard deviation method as in Fig 4. Precision values are the same as shown in Fig 6).
(EPS)

**S3 Fig. Estimates of $I_{d,sh}$ from the discrete NSB method with shuffling bias correction for all moth and muscle pairs.** Log plots of $I_{d,sh}$ at different values of $r_d$ for all moths and muscle types. Markers indicate precision values for each $I_{d,sh}$ curve found using a peak-finding method. Motor output states discretized to $b_m = 3$ output states per PC score (9 total motor output states).
(EPS)

**S4 Fig. Examples of discrete method with and without shuffling bias correction ($I_d$ and $I_{d,sh}$) for different motor output discretizations.** Log plots of $I_d$ (solid) and $I_{d,sh}$ (dashed) at different values of $r_d$ for one random example moth from each muscle type. Colors indicate total number of motor output bins $b_m^2$ used, and dots indicate spike timing precision estimate observed as peak $I_{d,sh}$.
(EPS)

**S5 Fig. Comparison of three different methods for selecting precision from mutual information drop-off with noise amplitude $r_c$. (A)** STD method, used throughout text, where $r_c$ is selected as the noise level where $I_c$ drops below the lower bound on the estimate of MI (mean minus one standard deviation) at $r_c = 0$. **(B)** Derivative method, where precision is found as the noise level where the peak of the 2nd derivative of $I_c$ with respect to $r_c$, normalized to an arbitrary scale, occurs. Peak finding is performed with a minimum amplitude requirement, depicted as the dashed line, as well as a prominence requirement. **(C)** Two-line intersection method, where lines are fit to approximately linear sections of the $I_c$ curve, defined as the 30 first and last noise levels. Precision $r_c$ is defined as the noise level where these lines intersect. **(D-F)** Estimates of precision on synthetic data comprised of a $1 \times N$ vector of "spike times" related to a $2 \times N$ matrix of "motor PC scores" via bivariate gaussians with fixed levels of precision (top panels) and on the main data of this paper with spike times fixed to ground-truth levels of precision (bottom panel). See Fig 5 for more details. **(G)** Mean ± one standard deviation for precision values from each method for all moth and muscle pairs, raw data displayed with jittered black dots. Red asterisks indicate distribution of precision estimates for a given method is significantly different ($p < 0.05$) from the STD method for that muscle in a two-sample $t$-test. The derivative method significantly deviates from the STD method for precision estimates on the AX and SA muscles ($p = 0.0363$ and $p = 0.0265$, respectively), while the two-line intersection method deviates from the STD method on estimates for the SA, DVM, and DLM ($p = 0.0303$, $p = 0.0013$, and $p = 0.0243$, respectively).
(EPS)

**S6 Fig. Nearest neighbor distance distributions as noise is added. (A)** Example of k-nearest neighbor distance distributions from a single moth and muscle pair (moth 3, RSA). Each subplot shows max-norm distance distributions for all data where a given number of spikes occurred in a wing stroke, with vertical lines indicating the median for each distribution. Max-norm distances were computed following the same algorithm of Kraskov [32] with $k = 4$. Note

that all quantities are standard scored before distance comparisons are made, so the x-axes for (A) and (B) are in units of standard deviations.

**(B)** Medians of $k = 4^{th}$ nearest neighbor distances as a function of noise added for an example moth and muscle pair (same as in (A)).

**(C)** Difference between medians of $k = 4^{th}$ nearest neighbor distances and median distance at zero noise, as a function of noise for all moth and muscle pairs. Stacked bar plots next to each plot show the proportion of wing strokes observed with each spike count for that muscle. Individual lines correspond to different observed spike counts, colored according to the legend in panel (D) (so black, for instance, is the k-NN distances for all wingstrokes where 1 spike was observed). Red dashed line with smaller markers indicates the mean shift in median k-NN distance weighted by the probability of each spike count. A y-axis value of zero indicates that the median nearest-neighbor distance at a given noise level is unchanged from the zero noise case. Error bars denote mean ± 1 standard deviation to show distribution of medians across different moths at each level of noise added for a given muscle. If an amount of spikes only occurred in 1 moth, dashed lines and unfilled circles are drawn as no mean or standard deviation can be calculated. Note that after distance comparisons are performed in the $|z - z'|$ space, standard scoring is removed so that (C) and (D) display distances in milliseconds rather than standard deviations.

**(D)** Difference between medians of $k = 4^{th}$ nearest neighbor distances and median distance at zero noise for all moth and muscle pairs. As in (C) distances are converted to milliseconds by multiplying by standard deviations of spike times to remove standard scoring.
(EPS)

**S7 Fig. Precision found by KSG estimator as k is varied.** Precision found by KSG estimator using the STD selection method (see S5 Fig) for all data (all moths included) of each muscle type as $k$ is varied from $k = 2$ through $k = 7$. Black jittered points show underlying data, with blue error bars denoting mean ± 1 standard deviation. No statistically significant differences are observed at any differing values of $k$.
(EPS)

**S1 Text. Procedure for estimation of spike timing precision using three separate algorithms.** Discussion of the reasoning for and underlying algorithms behind three separate methods to estimate spike timing precision from mutual information corrupted by uniform noise. Details on differences between methods and comparison between methods are discussed.
(PDF)

**S2 Text. Effects of noise and choice of k on nearest neighbor distances and the scale underlying mutual information estimation.** Discussion of the potential effects uniform noise could have on the nearest neighbor distance distributions of an underlying dataset in the process of estimating spike timing precision. Potential issues related to noise corruption and the choice of k, and their relevance to the precision estimation method, are detailed and evaluated.
(PDF)

## Author Contributions

**Conceptualization:** Joy Ortega, Simon Sponberg.

**Data curation:** Joy Ortega, Leo Wood.

**Formal analysis:** Joy Ortega, Tobias Niebur, Leo Wood, Rachel Conn, Simon Sponberg.

**Funding acquisition:** Simon Sponberg.

**Investigation:** Joy Ortega, Tobias Niebur, Leo Wood, Rachel Conn.

**Methodology:** Joy Ortega, Tobias Niebur, Leo Wood, Rachel Conn, Simon Sponberg.

**Project administration:** Simon Sponberg.

**Software:** Joy Ortega, Tobias Niebur, Leo Wood.

**Validation:** Joy Ortega, Tobias Niebur, Leo Wood.

**Visualization:** Joy Ortega, Tobias Niebur, Leo Wood.

**Writing – original draft:** Joy Ortega, Tobias Niebur, Simon Sponberg.

**Writing – review & editing:** Joy Ortega, Leo Wood, Rachel Conn, Simon Sponberg.

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
