## [Decision Letter · Decision Letter 0]

10 May 2023

Dear Dr. Sponberg,

We are pleased to inform you that your manuscript 'An information theoretic method to resolve millisecond-scale spike timing precision in a comprehensive motor program' has been provisionally accepted for publication in PLOS Computational Biology.

Best regards,

Peter E. Latham

Academic Editor

PLOS Computational Biology

Marieke van Vugt

Section Editor

PLOS Computational Biology

Reviewer's Responses to Questions

**Comments to the Authors:**

Reviewer #1: I thank the authors for their dedication to providing detailed additional explorations in response to my comments. I agree that this has provided excellent evidence for the conclusions, addressing my concerns about potential confounds. Indeed this is an exemplary response, which demonstrated solid understanding of the comments and sought to properly address confounds rather than sidestepping them. The new results truly underline the performance of this estimator. In particular, the inclusion of bias correction for the NSB estimator makes the improvement afforded by your estimator much more clear, as do the inclusion of synthetic data in response to reviewers 1 and 2. To my mind, the concerns of the other reviewers are well-addressed also.

So I'm happy to recommend acceptance, and again congratulate the authors on a job well done.

If the authors have a chance to make minor changes before publication, they could add the following (I don't need to check these):

1. After line 283 or so (and again at line 461), you might want to emphasise that nevertheless the resolution determined by the kth nearest neighbor distance is below the precision identified by the noise-addition method, which suggests that your estimate does not appear to be data-limited here. You've got a extended discussion on that in SI, and I think it would be useful for the reader for the main text to connect to / refer into that.

2. Fig S4 - is a bit small and difficult to make out, e.g. it's difficult to see which lines are dashed (though it looks to be the lower set in all cases). Perhaps the x axis can be zoomed in (much of it is unused) and/or the figures made larger.

3. Minor:

- Abstract: "In behaviors from slow breathing to rapid flight, motor use precise spike timing," seems garbled.

- line 45 "elctromyography"

Reviewer #2: The authors have addressee all of my concerns to my satisfaction. In particular, the addition of the analysis surrounding figure 5 have substantially improved the manuscript. Congratulations!

**Have the authors made all data and (if applicable) computational code underlying the findings in their manuscript fully available?**

Reviewer #1: Yes

Reviewer #2: Yes

PLOS authors have the option to publish the peer review history of their article (what does this mean?). If published, this will include your full peer review and any attached files.

Reviewer #1: **Yes: **Joseph Lizier

Reviewer #2: No

---

## [Editor Report · Acceptance letter]

6 Jun 2023

PCOMPBIOL-D-23-00478 

An information theoretic method to resolve millisecond-scale spike timing precision in a comprehensive motor program

Dear Dr Sponberg,

I am pleased to inform you that your manuscript has been formally accepted for publication in PLOS Computational Biology. Your manuscript is now with our production department and you will be notified of the publication date in due course.

With kind regards,

Zsofi Zombor
